# Cell Chromatography: Biocompatible Chromatographic Separation and Interrogation of Microbial Cells

M. Hazu,[a]* A. Ahmed,[a] E. Curry,[a]§ D. P. Hornby,[a] ⓘD. T. Gjerde[b]

[a]School of Biosciences, Firth Court, University of Sheffield, Sheffield, United Kingdom
[b]GjerdeTechnologies, Saratoga, California, USA

**ABSTRACT** The isolation of pure, single colonies lies at the heart of experimental microbiology. However, a microbial colony typically contains around 1 million cells at all stages of the life cycle. Here, we describe a novel cell chromatography method that facilitates the capture, purification, and interrogation of microbial cell populations from both single and mixed cultures. The method described relies on, but is not limited to, differences in surface charge to separate bacterial strains. The method is fully biocompatible, leading to no significant loss of cell viability. The chromatographic capture of cells, combined with selective elution methods, facilitates a greater level of experimental control over the sample inputs required for downstream high-throughput and high-sensitivity analytical methods. The application of the method for interrogating the antibiotic resistance of bacterial strains and for the separation of bacteria from environmental samples is illustrated.

**IMPORTANCE** This is the first report of a method for separating microbial cells using chromatography, with full retention of cell viability. Differences in the surface chemistry of microbial cells provides a means of attracting cells to immobilized microbeads. Some cells are attracted, and some are repelled. The differences in, for example, surface charge can be harnessed to capture, interrogate, and separate environmental samples, thus circumventing the need to use conventional bacterial plating methods. This method will greatly facilitate drug discovery and bioprospecting for novel microbial compounds.

**KEYWORDS** chromatography, cell chromatography, microbes, separation

The isolation of single colonies using selective growth medium has been the mainstay of microbiology since the development of the petri dish over 100 years ago. However, each colony of any pure bacterial strain is intrinsically heterogeneous with respect to its physiological growth status. In most cases, single colonies are initially obtained by subculturing in the laboratory from a petri dish or broth culture, or from "the field," via a clinical or environmental sample. Following some form of streaking or spreading protocol, combined with an empirical dilution of the sample, single colonies are typically obtained overnight (depending on the growth kinetics of the strain under investigation and the suitability of the growth medium). A single colony will typically contain around 1 million cells, all of which will be at a different stage in the life cycle; at extremes, some will be newly replicated, while others will be dead. However, as molecular analysis techniques become increasingly powerful, higher in throughput, and capable of single cell resolution, the traditional microbiology methods for both the analytical and preparative purification of cells have become limiting.

Liquid chromatography has been one of the most successful and enduring methods for the purification of biological molecules since its widespread uptake by biochemists 60 years ago (see, for example, Duong-Ly and Gabelli [1]). Many commercial suppliers of chromatography media and instruments have focused on the separation of proteins, peptides, and nucleic acids, where a wide range of protocols

Address correspondence to D. T. Gjerde, doug.gjerde@outlook.com.

*Present address: M. Hazu, Division of Biology and Biological Engineering, California Institute of Technology, Pasadena, California, USA.

§Present address: E. Curry, Department of Biological and Chemical Engineering, University of Sheffield, Sheffield, United Kingdom.

The authors declare no conflict of interest.

**TABLE 1** Estimation of the impact of cell chromatography on cell viability[a]

| Characteristic | Data for: | |
| --- | --- | --- |
| | Sample | Replicate |
| No. of colonies with no chromatography | 1,304 | 1,258 |
| No. of colonies postchromatography | 1,239 | 1,185 |
| % viability | 95 | 94 |

[a]Equivalent serial dilutions were made of cells before and after uploading and elution of *E. coli* BL21(DE3) from two independent experiments. This protocol was incorporated as a control for each of the chromatography experiments described, and the viability was never less than 90%.

and chromatographic media have been developed which combine high resolution with retention of biochemical function. In parallel, chromatography resins and instruments have been developed to support downstream chemical analysis (mainly peptide sequence determination) where functional integrity is not required. Mass spectrometry of proteins and peptides is probably the best example of a technology that is dependent on the chemical, but not functional, integrity of input samples. In contrast, biochemical assays and structural biology techniques, including X-ray crystallography, nuclear magnetic resonance (NMR) spectroscopy, and cryo-electron microscopy, all demand both high levels of purity and functional integrity of the sample.

As microscopic techniques for interrogating cells and tissues approach the resolution of macromolecular structure determination methods reviewed by Thorn (2), the demand for reproducible cell purification methods is becoming increasingly important. The first successful application of liquid chromatography for the separation of microbial cells was described by Arvidsson et al. (3), who used supermacroporous cryogels for the separation of bacteria. They showed that ion exchange and immobilized metal ion chromatography can be used to separate microbial cell mixtures with moderate retention of cell viability. Using conventional liquid chromatography protocols, bacterial cells are applied to a cryogel column and are captured at low ionic strength. The captured cells are subsequently eluted following methods that are commonly applied to protein separation.

Here, we describe a novel method for the purification of cell populations using ion-exchange chromatography, although there is no *a priori* limit to the choice of stationary-phase medium. However, unlike conventional chromatography, cells and buffers enter and exit the column in a back-and-forth flow (4, 5). We refer to this method simply as "cell chromatography." This method has been applied to the separation of a mixture of Gram-positive (*Staphylococcus aureus*) and Gram-negative (*Escherichia coli*) bacteria. In addition, we demonstrate the resolving power of the method for the separation of individual strains found in local environmental water samples. Just as proteins in complex cell extracts can be separated by ion-exchange chromatography, the differences in net surface charge are sufficient to separate heterogeneous populations of cells, with no significant loss of viability.

## RESULTS

**Quantifying capture and release of bacteria via cell chromatography.** Throughout all the experiments reported here, a single protocol was used, in which either broth cultures or liquid field samples were applied using a regulated back-and-forth flow through an ion exchange resin packed within conventional 1-mL disposable pipette tips.

The chromatographic retention and the viability of cells following cell chromatography are exemplified in Table 1 using the common laboratory strain *E. coli* BL21(DE3). There was occasionally a small reduction in CFU observed as a result of the separation, but this loss of viability never exceeded 10%. Similar results were obtained for several other laboratory strains. The results in Tables 2 to 5 show that approximately 5 to 10% of cells from a given broth culture are captured by the columns in the format supplied. In most

**TABLE 2** Estimation of the fraction of *E. coli* cells ($3 \times 10^5$ uploaded in each case) bound to individual columns

| Data | Value for: | | | |
| --- | --- | --- | --- | --- |
| | Fraction 1 | Replicate 1 | Replicate 2 | Replicate 3 |
| No. of colonies at (mM of NaCl): | | | | |
| 0 (25th wash fraction) | 984 | 804 | 863 | 973 |
| 50 | 3,704 | 1,984 | 2,345 | 2,764 |
| 100 | 4,128 | 3,682 | 4,032 | 4,256 |
| 150 | 5,728 | 4,893 | 5,409 | 5,610 |
| 200 | 5,936 | 4,690 | 5,537 | 6,109 |
| 300 | 4,902 | 5,783 | 4,723 | 5,267 |
| 400 | 1,928 | 1,423 | 1,746 | 2,105 |
| 500 | 1,112 | 812 | 1,092 | 1,273 |
| 750 | 416 | 444 | 321 | 492 |
| 1,000 | 146 | 128 | 113 | 182 |
| Total no. of colonies | 28,000 | 23,839 | 25,318 | 28,058 |
| % cells bound[a] | 9.33 | 7.95 | 8.44 | 9.35 |

[a]Determined by serial dilution and standard CFU scoring on agar plates.

experiments, the capacity of the columns is compatible with the requirements of downstream analysis, in this case using agar plates to determine the outcome of a particular experiment. The volume of chromatography medium can be increased (or decreased) to fit a particular application; for example, there was no significant difference in performance, other than an increase in cell retention, using larger bed volumes of up to several hundred microliters (unpublished data).

**Separation of bacterial cells by cell chromatography.** The primary aim of this work was to determine whether the cell surface characteristics of living cells could be exploited to separate bacterial cells in a manner analogous to ion-exchange chromatography of proteins. Separate broth cultures of *E. coli* BL21 (carrying pET22b, an ampicillin-resistant plasmid) and *S. aureus* (SH1000) were applied to ion exchange tips, as described in Materials and Methods. The results of a stepwise application of NaCl as the eluting agent are shown diagrammatically in Fig. 1A. At pH 7.4, *E. coli* and *S. aureus* strains can be adequately resolved, with some degree of overlap. Pure cultures can readily be obtained by taking early eluting fractions from the *E. coli* profile, or late eluting fractions from the *S. aureus* cells. These data are very similar to those obtained when separating two polypeptides differing in net surface charge at a given pH (6). Further optimization can be achieved by altering the pH and by controlling the stepwise addition of elution agent, in this case NaCl, as with any ion exchange

**TABLE 3** Estimation of the fraction of *E. coli* cells ($5 \times 10^5$ uploaded in each case) bound to individual columns

| Data | Value for: | | | |
| --- | --- | --- | --- | --- |
| | Fraction 1 | Replicate 1 | Replicate 2 | Replicate 3 |
| No. of colonies at (mM of NaCl): | | | | |
| 0 (25th wash fraction) | 213 | 173 | 167 | 193 |
| 50 | 693 | 478 | 435 | 528 |
| 100 | 1,363 | 1,223 | 1,290 | 1,309 |
| 150 | 1,765 | 1,545 | 1,368 | 1,573 |
| 200 | 1,492 | 1,145 | 1,092 | 1,390 |
| 300 | 579 | 431 | 675 | 545 |
| 400 | 331 | 359 | 202 | 298 |
| 500 | 132 | 123 | 98 | 105 |
| 750 | 108 | 93 | 65 | 43 |
| 1,000 | 27 | 83 | 40 | 7 |
| Total no. of colonies | 6,190 | 5,478 | 5,265 | 5,798 |
| % cells bound[a] | 12.38 | 10.96 | 10.53 | 11.59 |

[a]Determined by serial dilution and standard CFU scoring on agar plates.

**TABLE 4** Estimation of the fraction of *S. aureus* cells ($3 \times 10^5$ uploaded in each case) bound to individual columns

| Data | Value for: | | | |
|---|---|---|---|---|
| | Fraction 1 | Replicate 1 | Replicate 2 | Replicate 3 |
| No. of colonies at (mM of NaCl): | | | | |
| 0 (25th wash fraction) | 444 | 509 | 549 | 465 |
| 50 | 2,965 | 2,305 | 2,789 | 2,340 |
| 100 | 3,520 | 3,109 | 3,356 | 3,271 |
| 150 | 4,509 | 3,756 | 4,137 | 3,987 |
| 200 | 3,229 | 3,912 | 3,547 | 3,730 |
| 300 | 1,732 | 1,907 | 1,809 | 1,869 |
| 400 | 902 | 933 | 810 | 758 |
| 500 | 630 | 739 | 548 | 586 |
| 750 | 214 | 312 | 253 | 201 |
| 1,000 | 89 | 119 | 133 | 79 |
| Total no. of colonies | 17,790 | 17,092 | 17,382 | 16,821 |
| % cells bound[a] | 5.93 | 5.70 | 5.79 | 5.61 |

[a]Determined by serial dilution and standard CFU scoring on agar plates.

procedure. Moreover, a second dimension of separation can be achieved by using any suitable affinity resin, or a cationic resin.

In order to demonstrate that bacteria from mixed cultures can be separated by cell chromatography, two broth cultures of *E. coli* and *S. aureus* were mixed and the combined culture uploaded onto an ion exchange column. Following chromatography and collection of the salt-eluted material, visual inspection of the agar plates demonstrated that the two strains eluted at the expected salt concentrations, as observed when individual cultures were applied. For clarity, the experiment was repeated with *E. coli* BL21, harboring an ampicillin resistance plasmid. This makes it possible to compare the eluted fractions on Luria agar (LA) plates with and without ampicillin. As can be seen in Fig. 2, the two strains are clearly separated by a stepwise addition of sodium chloride solutions of increasing concentrations.

**Separation of environmental strains by cell chromatography.** The emergence of the capabilities for microbiological and genomic analysis of unculturable microbes represents a major new opportunity for bioprospecting. Historically, the recovery of culturable species has been the focus of taxonomic and pharmaceutical/biotechnological applications, and access to hitherto unexploited genes and metabolites represents a major shift in these fields. In Fig. 3, a series of strains isolated from a local pond were uploaded as a

**TABLE 5** Estimation of the fraction of *S. aureus* cells ($5 \times 10^5$ uploaded in each case) bound to individual columns

| Data | Value for: | | | |
|---|---|---|---|---|
| | Fraction 1 | Replicate 1 | Replicate 2 | Replicate 3 |
| No. of colonies at (mM of NaCl): | | | | |
| 0 (25th wash fraction) | 145 | 136 | 146 | 123 |
| 50 | 489 | 462 | 507 | 424 |
| 100 | 798 | 875 | 806 | 693 |
| 150 | 1,232 | 1,156 | 1,178 | 1,098 |
| 200 | 920 | 815 | 904 | 722 |
| 300 | 789 | 642 | 736 | 634 |
| 400 | 334 | 309 | 312 | 298 |
| 500 | 145 | 136 | 176 | 117 |
| 750 | 98 | 78 | 102 | 86 |
| 1,000 | 46 | 67 | 56 | 60 |
| Total no. of colonies | 4,851 | 4,540 | 4,777 | 4,132 |
| % cells bound[a] | 9.70 | 9.08 | 9.56 | 8.23 |

[a]Determined by serial dilution and standard CFU scoring on agar plates.

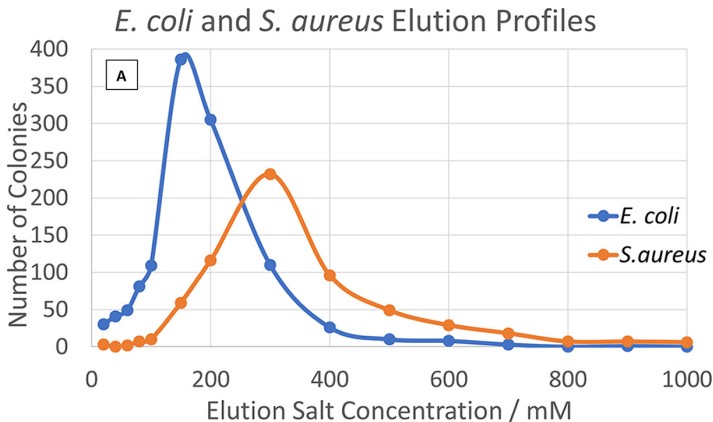

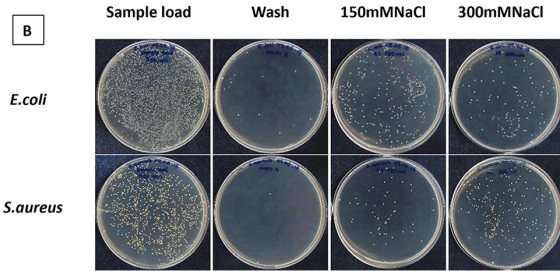

**FIG 1** (A) Separation of *E. coli* and *S. aureus* cells by cell ion-exchange chromatography. In the example shown, two liquid cultures of cells were mixed and uploaded onto an ion exchange tip. CFU were determined by serial dilution and plating. (B) Petri dishes showing the growth of samples from eluted fractions; *E. coli* and *S. aureus* broth cultures were independently applied to a cryogel column, followed by stepwise salt elution. The cross contamination of cells at 150 mM NaCl is consistent with the elution profile shown in panel A.

mixture onto a cell chromatography column. As before, a controlled application of a stepped gradient of NaCl led to a clear resolution of some of the species in the mixture. For simplicity, strains were selected for their clear pigmentation differences, as a proof of principle for the technology. The remarkable power of resolution of cell chromatography is shown in Fig. 3 (top), where the differential elution of the pigmented microorganisms can be clearly observed. In this experiment, an unknown set of culturable strains were separated. However, the downstream plate cultures serve to demonstrate that separation of strains has been achieved. The eluted fractions will also contain nonculturable bacteria, which can be readily accessed for many molecular and whole-genome analyses.

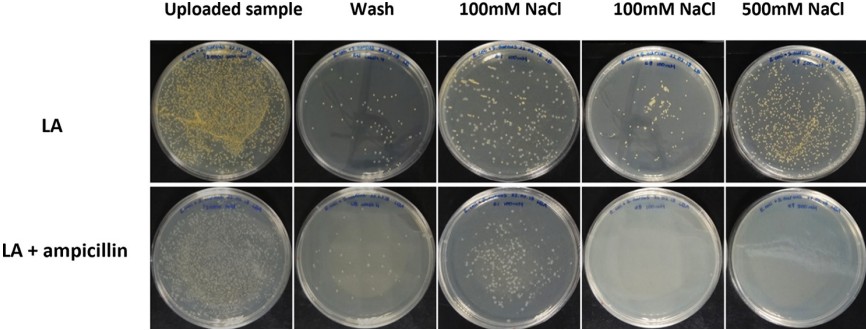

**FIG 2** Separation of *E. coli* and *S. aureus* cells by cell chromatography. The upper series of plates shows the growth of fractions eluted from a column and plated onto LA in the absence of antibiotics, where both strains grew well. The lower gallery shows the growth of the same fractions spread onto ampicillin-containing LA plates; only antibiotic-resistant *E. coli* cells were recovered, clearly indicating that effective separation was achieved.

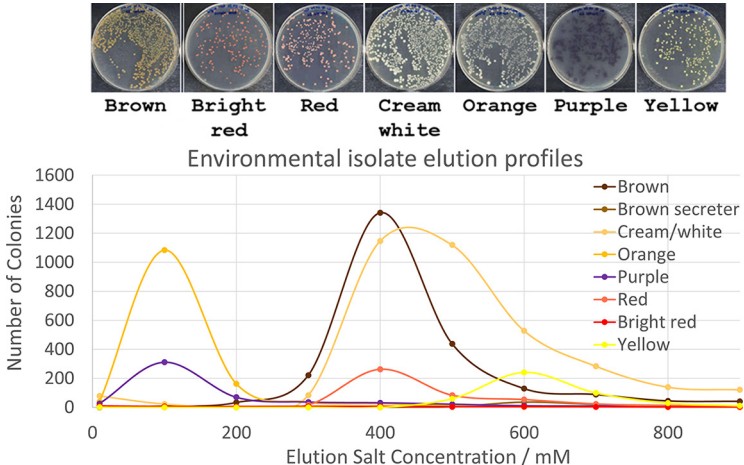

**FIG 3** Elution profiles of a mixture of pigmented strains isolated from local pond water. A 1-mL sample was uploaded directly onto a column, and as in Fig. 1, the eluted fractions were diluted and the CFU counted from petri dishes, as shown in the gallery (top).

**Interrogation of immobilized bacterial cells with antibiotics.** The search for novel antibiotics is a global scientific and medical priority. Ever since the serendipitous discovery of penicillin over 90 years ago, the technology used to screen antimicrobial candidates still relies on the combination of petri dishes, solid media, and cellulose disks or strips impregnated with a range of antibiotics, often at a range of concentrations. Contact with potential antibiotics indicated by a zone of growth inhibition (see reference 7) typically provides the first indication of antibiotic sensitivity.

*E. coli* BL21(DE3) cells harboring a plasmid conferring ampicillin resistance (pET22b) were combined with an equivalent number of *S. aureus* SH1000 cells. The mixed culture was uploaded onto an ion exchange resin as described previously, followed by a number of washes with LB medium and a final short (10 s) incubation of immobilized cells in LB supplemented with ampicillin at a concentration of 200 $\mu$g/mL. Cells were subsequently eluted at a series of increasing concentrations of sodium chloride. Samples from all eluted fractions were subsequently plated out and incubated for 16 h on antibiotic-free Luria agar at 37°C. As can be clearly seen from Table 6, this protocol mimics the outcome of a typical plating experiment; there is an approximately 50-fold reduction in cell numbers where the strain is ampicillin sensitive. Clearly, manipulation of the incubation periods, cell numbers, and antibiotic concentrations would form the basis of a more thorough "interrogation," but nonetheless, it is clear that the cell chromatography protocols described here can replace traditional petri dish-based experiments in some of the fundamental procedures used in experimental microbiology. Moreover, these chromatographic protocols are much more suited to automation.

## DISCUSSION

Attempts to purify living bacterial cells for both analytical and preparative purposes in broth cultures have proved challenging, with most resins being utilized in a batch

**TABLE 6** Results of cell purification protocol using liquid chromatography[a]

| Bacterial strain | No. of colonies in: | | |
| --- | --- | --- | --- |
|  | Replicate 1 | Replicate 2 | Replicate 3 |
| *E. coli* BL21 | 824 | 922 | 809 |
| *S. aureus* SH1000 | 12 | 21 | 15 |

[a]0.5 mL of a suspension comprising ~25,000 BL21 cells and ~25,000 SH1000 cells per mL was uploaded, followed by 25 wash steps to remove the majority of unbound bacteria. 0.5 mL of LB broth containing 100 $\mu$g/mL ampicillin was then applied to the column and all flow stopped for 10 s, after which the column was flushed out with LB. Cells were subsequently eluted from the column using 0.5 mL LB broth containing 200 mM NaCl, and the resulting eluent was plated onto LB agar plates as before.

mode, using centrifugation or magnetism to recover and interrogate resin-bound cells. While it is clearly possible to capture and separate cells that have been "fixed" in some way, such methods provide limited insight into cellular physiology. In 2006, Arvidsson et al. were the first to demonstrate that cryogels could be used as a stationary phase for the capture and separation of viable bacteria using conventional unidirectional chromatography. Clearly, there is no *a priori* barrier to cell chromatography. Here, we have utilized a novel form of capture and purification, introducing a back-and-forth mode of sample application and elution that simplifies the process and paves the way for a more controlled approach to cell analysis that goes some way to meeting the needs of high-throughput, high sensitivity "omics" methods.

A population of microbial cells in an asynchronous culture (a typical batch culture) show a level of morphological diversity that is not dissimilar in principle to the diversity of polypeptides expressed in a cell. A microbial cell, or a virus particle, can be considered a "studded" sphere (or cylinder), with the net charge distribution falling on a wide scale from negative through neutral to positive. In comparison with proteins, the considerably larger surface area of cells and viruses (or particles in general), as well as the particle size, must be considered if retention of biological competence is required. In addition to ion exchange media, chromatography resins coupled to affinity ligands, including small molecules and antibodies, can provide a biocompatible surface for the selective purification of cells from mixed populations, thereby enriching for a specific subpopulation. Such beads may be magnetic and are often incorporated into a centrifugation-associated protocol. To date, however, there are very few examples of the successful chromatographic separation of cells and viruses, where the biological specimen retains full biological viability.

The column capacity of an individual column for bacteria is related to the ion exchange capacity and the equilibrium constant of the bacteria and anions in the buffer competing for the ion exchange sites. If the bacteria compete weakly, then the column capacity is low. However, if the bacteria compete strongly, then the column capacity is high, owing to the multivalent attachment of the bacteria on the surface of the anion exchanger, where the selectivity of the bacteria is normally high but the kinetics of capture are relatively slow. There are several rate constants that contribute to the overall rate of capture, including those describing the binding of the bacteria to the anion exchange functional groups and the orientation of the anionic sites of the bacteria to the positive charges bound to the resin. Finally, capture of bacteria is likely to be via multipoint attachment. Since multipoint attachment increases the selectivity of binding, time is needed to maximize the attachment of a particular bacterium to the resin bead. This strong attachment is supported by back-and-forth flow through the column, which gives multiple opportunities for bringing the bacterium to the ion exchange site, thereby optimizing the orientation of positive and negative sites and the strongest multipoint attachment of the bacterium. However, back-and-forth flow is *a priori* deleterious to cells, damaging or killing them because of multiple chances of puncture, trapping, or shearing. The columns have been designed to minimize these possible harmful interactions.

The selectivity of bacteria for ion exchangers is described by sharp isotherms, as shown by the results presented in this paper. This means that for any particular type of bacterial cell, of a given density and net negative charges, and for any given set of buffer conditions, the bacteria are either mostly bound to the resin or mostly in solution. Furthermore, as the buffer conditions are altered, there is a sharp transition of buffers where there is sorption of the bacteria to the resin compared to conditions where there is nonsorption of the bacteria to the resin. Thus, small changes in the mobile phase buffer can result in a significant impact on whether a particular bacterium adheres to the column or does not. In other words, multipoint attachment makes the isotherms related to ion exchange extremely sharp, and small changes in buffer conditions (ion type and concentration) can result in large shifts in interaction affinities. In this work, a set of conditions were chosen to capture all bacteria of

different types. Thereafter, the concentration of ions competing for the ion exchange sites was increased. This resulted in bacteria being released from the beads and eluted from the column. Since different bacteria have different selectivity for the anion exchanger, separation of the bacterial types was accomplished.

Until relatively recently, environmental microbiologists necessarily focused their attention on the characterization of culturable microorganisms. However, many microscopically observed species remained just that until the arrival of direct genome sequencing (8). The introduction of cell chromatography for the capture and interrogation of live cells in a simple chromatographic format has the potential to transform the systematic analysis of many fundamental properties of bacteria. In addition, both bioprospecting and antimicrobial discovery programs should benefit significantly from the potential for automation afforded by this technology.

The sensitivity of this method requires further detailed investigation, since the environmental sample analysis described here is likely to be dominated by the most abundant species present. However, the method can be readily scaled to maximize the capture of organisms present in low abundance (unpublished results). Experiments are currently in hand to explore the composition of microbiome populations, which combined with parallel genome sequence analysis should provide an invaluable strategy for both fundamental research and bioprospecting.

## MATERIALS AND METHODS

**Growth and isolation of bacterial strains.** *Staphylococcus aureus* SH100 (a kind gift from Simon Foster, The University of Sheffield) and *Escherichia coli* BL21(DE3) (with or without prior transformation with the expression vector pET22b [Novagen]) cells were cultured overnight at 37°C on Luria broth (with or without agar) and supplemented with 200 $\mu$g/mL ampicillin (Sigma) as appropriate. Environmental samples of pond water (10 mL) were collected in sterile tubes from a local boating lake adjoining the University of Sheffield and were used without further culturing, on the same day, with or without dilution into phosphate-buffered saline (PBS) at pH 7.4. All standard methods were carried out as described by Sambrook et al. (9).

**Chromatography.** Pipette tip-based columns provided by Gjerde Technologies contained a strong base anion exchanger bound to a water-swollen agarose substrate. The columns were packed without compression to retain flow paths that would be nonrestrictive to bacterial cells. Pipette tip columns containing 100 $\mu$L of Cytiva Q Sepharose Fast Flow anion exchange resin (average particle size, 90 $\mu$m) in a 1-mL pipette tip with 0.635-cm diameter frits were placed firmly onto an automatic pipette, ensuring a tight seal to allow for efficient and reproducible uptake of liquids. Each column was first equilibrated by withdrawing 500 $\mu$L of PBS solution, pH 7.4, at a rate of 0.75 mL/min from a 96-well plate or Eppendorf tube; then, after 10 s, 400 $\mu$L of the PBS solution was ejected. Care was taken to ensure that the column did not dry out; the total volume of liquid in the column was maintained above 100 $\mu$L. The total volume of the resin was ~100 $\mu$l. The tips were mounted on a 6-channel, semiautomatic pipette, and the flow rates and directions were driven by automated pipette software. For the work performed here, Rainin E4 pipettes equipped with PureSpeed software (Biotage, San Jose, CA) were used to control the back-and-forth flow.

Typical bacterial suspensions were generated, containing between 104 and 105 cells/mL of Luria broth. These cell densities could be reproducibly and conveniently counted on agar plates following elution. In a typical protocol, 500 $\mu$L of a bacterial suspension was withdrawn from a liquid culture at a rate of 0.75 mL/min, bringing the total column volume to 600 $\mu$L. After 10 s, 500 $\mu$L of the column contents was ejected at the same rate, releasing the majority of unbound bacteria.

In order to remove any bacteria that had bound nonspecifically to the column matrix or other areas of the column, such as the exterior of the tip, the column was washed with 25 consecutive column volumes of LB broth. The bound fraction of cells was then eluted with 500 $\mu$L LB broth into an Eppendorf tube and the total volume adjusted to 1 mL with LB broth (for immediate use) or LB supplemented with glycerol (final concentration, 25% [vol/vol]) if samples were to be frozen. When required, in order to prevent further growth during chromatography, the samples were washed with sterile PBS, without any significant loss of cell viability.

Elution of the bound cells was achieved as follows. After 25 consecutive LB (or PBS) washes, matrix-bound cells were eluted by the back-and-forth addition of either LB or PBS containing increasing concentrations of NaCl from 50 mM to 1 M. At each NaCl concentration, 500 $\mu$L LB broth (or PBS), containing the required concentration of NaCl, from a total volume of 1 mL in an Eppendorf tube, was applied at a rate of 0.75 mL/min and then immediately ejected. The fractions were either analyzed immediately or frozen in order to minimize any further growth of the cells before plating. The whole process took approximately 2 h to complete (time would vary depending on the sample volume and the number of wash and elute cycles required).

The eluted bacterial fractions were then plated onto LB agar (with or without ampicillin) and incubated for 16 h at 37°C. The bacterial cell numbers (CFU) were determined by serial dilution of the broth

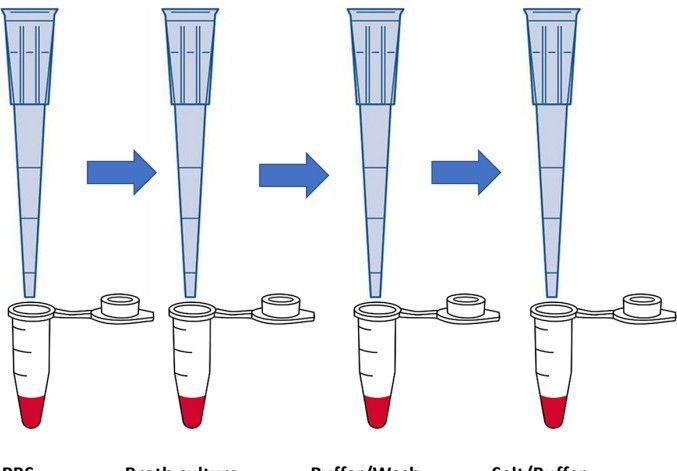

**1.Equilibrate column    2. Upload cells         3.Wash (25x)       4. Step-wise elution**

**PBS            Broth culture        Buffer/Wash        Salt/Buffer**

**FIG 4** General illustration of the cell chromatography columns in a tip format, suitable for use with most commercial semiautomatic or fully automated air displacement pipettes. In step 1, the column is equilibrated by back-and-forth flow of the chosen buffer. In step 2, cells are "uploaded" from a broth culture or field sample, followed by step 3, extensive back-and-forth washing (25×) with loading buffer. Finally, in step 4, cells are eluted by stepwise additions of (in this case) PBS supplemented with NaCl.

cultures, followed by agar plating. All experiments were carried out in triplicate unless otherwise indicated. The arrangement of the pipette tip chromatography columns is shown in Fig. 4.

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
