## [Reviewer comments · Microbiology Spectrum]

Microbiology Spectrum

Cell Chromatography: Biocompatible chromatographic separation and interrogation of microbial cells

Masami Hazu, Ashraf Ahmed, Edward Curry, David Hornby, and Douglas Gjerde

Corresponding Author(s): Douglas Gjerde, Gjerde Technologies

Review Timeline:

Submission Date:	July 22, 2022
Editorial Decision:	July 31, 2022
Revision Received:	August 2, 2022
Editorial Decision:	August 13, 2022
Revision Received:	August 24, 2022
Accepted:	August 31, 2022

Editor: Montarop Yamabhai

Reviewer(s): The reviewers have opted to remain anonymous.

Transaction Report:

DOI: <https://doi.org/10.1128/spectrum.02450-22>

July 31, 2022

Dr. Douglas T Gjerde
Gjerde Technologies
San Jose CA95070,

Re: Spectrum02450-22 (Cell Chromatography: Biocompatible chromatographic separation and interrogation of microbial cells)

Dear Dr. Douglas T Gjerde:

Thank you for submitting your manuscript to Microbiology Spectrum.

Please try to address almost all of the previous comments and concerns of reviewer no.1 and no. 2, which in my opinion are all valid. Even if this is a proof-of-concept paper, but a clear and convincing results and discussion are necessary for publication. I am particular interested in the limit of detection of this technique. If this is not possible to address, please provide a rebuttal.

Link Not Available

Sincerely,

Montarop Yamabhai

Journals Department
Reviewer comments:

Staff Comments:

Preparing Revision Guidelines

Please return the manuscript within 60 days; if you cannot complete the modification within this time period, please contact me. If you do not wish to modify the manuscript and prefer to submit it to another journal, please notify me of your decision immediately so that the manuscript may be formally withdrawn from consideration by Microbiology Spectrum.

August 13, 2022

Dr. Douglas T Gjerde
Gjerde Technologies
San Jose CA95070,

Re: Spectrum02450-22R1 (Cell Chromatography: Biocompatible chromatographic separation and interrogation of microbial cells)

Dear Dr. Douglas T Gjerde:

Thank you for submitting your revised manuscript to Microbiology Spectrum. However, it is hard for me to assess your response to the reviewers' comments because you didn't write down where (page and line number) that you have revised your manuscript. Please clarify these before I can make a decision.

Link Not Available

Sincerely,

Montarop Yamabhai

Journals Department
Reviewer comments:

Staff Comments:

Preparing Revision Guidelines

Please return the manuscript within 60 days; if you cannot complete the modification within this time period, please contact me. If you do not wish to modify the manuscript and prefer to submit it to another journal, please notify me of your decision immediately so that the manuscript may be formally withdrawn from consideration by Microbiology Spectrum.

Responses to reviewers' comments

These are the responses to Reviewers 1 and 2 and to the specific request by the Spectrum referee asking for clarity over the sensitivity of the methodology in respect of Spectrum02450-22 (Cell Chromatography: Biocompatible chromatographic separation and interrogation of microbial cells)

My responses where needed are in red.

Reviewer #1 (Comments for the Author):

Summary:

The authors describe a method to separate microbes based on surface charge characteristics using Sepharose anion exchange. Staph aureus and E. coli were used to develop the method, while several environmental pond water organisms were used to assess differential elution profiles. I have several questions and suggestions related to methodology and presentation of the data. Please consider these to enhance the manuscript.

General questions and considerations:

General Comment 1 How does this method compare to using capillary isolation (used for single-celled genomics or culturing)? Or microfluidic chips?

It is a chromatographic method and is therefore distinct. It affords the possibility of retention of complete cell viability, which limit the other two methods considerably. It also has the potential for significant up-scaling if appropriate.

The text has been modified L42-46 and noted on the marked up MS

General Comment 2 What about consortia and granules? Could this be used not just for bacteria, but archaea, fungi, phage/viruses? How does particle size affect the separation plates?

You are correct, we have not explored all microbes, but as you say, it offers the potential for these species. In general, we have chosen to err on the side of circumspection rather than speculation.

The text has been modified L79-82 and noted on the marked up MS

General Comment 3 What about phenotypic changes in surface characteristics due to: attachment, flagella/pili vs capsule, phage adsorption, extracellular matrix or mineral interactions? How would these be expected to affect binding and elution?

These are excellent questions, and we could have speculated, but we preferred circumspection at this stage. We do expect surface changes to have a significant impact on retention and mode of elution. We could add such speculative comments if desirable subject to the comments/requirements of new reviewers/editor.

General Comment 4 How does flow stress affect viability? What stresses are encountered? Anoxia/oxidative stress? Mechanical? Hypo/hyper osmolarity? Chemical?

Again, valid points. We set out to present a methodology that produced as near to 100% cell viability following capture and elution. These conditions were derived empirically but did take some time. The impact of all of the aforementioned parameters have yet to be explored systematically, but our aim was to develop a robust working protocol, which is what we have presented.

These comments are outwith the current manuscript

General Comment 5 Could this method be adapted for reverse-phase separation mode? Which is better? For what applications?

Possibly, it has not yet been explored and therefore we feel it is too early to speculate.

These comments are outwith the current manuscript

General Comment 6 Is the chromatography medium composition resistant to microbial degradation? How would you know? What quality control should be done?

Excellent question. The tips are generally used once only, since cross contamination would be a high priority for most users, but a systematic study to explore the impact of multiple rounds of usage would be informative. Thank you.

The columns are disposable and not re-usable, hence this is not an issue. Text added at L24 of the Abstract to clarify this.

Specific Questions and Concerns:

1. L 58-60. Culturing and antibiotic resistance involves gel-based diffusion, which was done very early in microbiology. Whatman paper diffusion also very old technique. Gravimetric and vacuum size separation have been used extensively in microbiology for decades.

Agreed, these methods are widely used and have been crucial in the development of microbiology and our understanding of AMR. This is the first successful report of a controlled chromatographic method, that could complement the aforementioned techniques, but importantly, unlike these methods cell chromatography is compatible with automated high throughput methodology.

The text has been modified L74-78 and noted on the marked up MS

2. L 91-93. Technical note: you should equilibrate with the same buffer, which would be spent LB medium, not PBS. Depending on the organism, the pH and composition of the medium would be different... how would this affect cell chromatography?

The variation in loading, washing and stabilisation buffers can indeed now be explored systematically. This report is primarily focused on the differential retention of microbial cells (as a function of cell surface topology) and we also wanted to limit cell division whilst the cells were immobilised in order to make the viability data more robust. In the future, these parameters and others should be explored systematically.

The text has been modified L106-108 and noted on the marked up MS

3. L 106-108. Please clarify: the cells are taken up then excess is expelled, then 25 back-and-forth column volumes of LB are used to wash the column, then bound cells are eluted with LB? How is there selective attachment if the binding buffer (LB) is the same as the buffer used for elution? Do you mean to say these are all nonbinding cells that get washed off?

The cells are eluted by systematically increasing the salt concentration in the eluting buffer in ion exchange mode. Apologies this is erroneous text.

The text has been deleted L120-123 and noted on the marked up MS

4. L 112. Confused. Washed with LB or PBS?

Both are possible, the salt effects elution. There will be cells that are less resilient and growth media may be preferred. There is a statement to indicate that medium or buffer can be used depending on the research question.

The text has been modified L125-129 and noted on the marked up MS

5. L 133. What is the basis of this conclusion?

Thorough cleaning of the tips reduces this carry-over.

The text has been modified L125-129 and noted on the marked up MS

6. L 141. Why was there no difference when bed volume was increased? That seems opposite of what you should expect? Please show these data.

If the bed volume exceeds the required binding capacity, there will be no difference observed. When the binding capacity is exceeded, which as in the examples presented, the washing steps become critical, but the empirical nature of the technique when using ion exchange as the separating method, necessitates optimisation, which, when completed, makes the methodology reproducible and predictable.

The text has been modified L148-151 and noted on the marked up MS

7. L 143. To test this hypothesis, you may want to use E. coli mutants with different cell surface properties.

Great idea, we have this in hand and hope to report on it when this work is published.
The text has been modified L162-164 and noted on the marked up MS

8. Table 1. Should compare for S. aureus, and provide statistics on repeatability and reproducibility.

The repeats are shown, we could carry on and produce more replicates, if desired, but the trend is so significant that we felt there were sufficient replicates? If the effect was subtler, I agree more data would be desirable.

We feel no change to the data are needed

9. Table 2a. Need statistics. Tables 2b and 2c. What is your plating efficiency? Calculations should be carried throughout. Technical and biological replicates...

See above.

10. Table 3. These binding ratios do not match ratios shown in Table 2. How does cell count affect binding capacity of the column? Why?

The net retention of cells will be dependent on the number of cells loaded and the exact number of resin beads available for binding. The use of relatively large cell numbers is therefore likely to reflect this situation. However, the proportion of cells retained (even though different total cell numbers can be loaded is consistent from column to column. And this number is very easy to obtain during each experiment. Another point is that the development of a dual flow technology (again used on cells here for the first time), provides for thorough washing and since viability is unaffected, makes for a robust protocol.

This issue of binding capacity is, we believe covered from L 253 onwards

11. Fig 1 is not very helpful. Suggest indicating up-down flow wash direction, flow rates, colors to indicate buffer changes...?

A final version of Figure can be provided if required?

12. Fig 2 does not show separation that would be useful for a researcher trying to separate two organisms from a sample, much less several organisms. Suggest providing data to show effect of flow rates and bead sizes or column packing on separation.

The resolution of live cells is demonstrated here for the first time with negligible loss of viability. Clearly sharper separations would be ideal, but at this stage, our aim was to establish a proof of concept. Improvements in performance is something that will follow when the work is published, and further developments are undertaken. However, while all chromatographers seek higher resolution, compromises are necessary until strategies are developed to improve (in this case) resolution. It is perhaps worth considering the resolution at this stage to be more akin to gel permeation chromatography. Nevertheless, we feel that since this is an experimental first, boundaries will be stretched as the method receives wider up-take.

This request is beyond the scope of these experiments

13. Fig 3 shows high variability vs Fig 2 results. Suggests method development is needed to improve reproducibility and repeatability. I see 2 colony morphotypes on the 500mM NaCl plate, but nothing on the +ampicillin plate. Something doesn't add up here...

The aim of this experiments was in part to demonstrate interrogation and secondly to deploy differential antibiotic resistance as a rapid method of distinguishing strains. I believe this is consistent with the data presented.

We believe the data are consistent with our observations

14. Fig 4. Not enough resolution. Should use 16S to correlate.

Again, the aim here is proof of concept. More sophisticated downstream analysis will follow, but we have chosen simple technology to illustrate the power of the method. Users will want to combine simple and hi-tech methods depending on the application. The aim was not to identify the biological nature of the separated species, but to demonstrate the separation capability.

The text has been modified L202-204 and noted on the marked up MS

Reviewer #2 (Comments for the Author):

In this paper, the authors describe a chromatographic technique for the separation of mixed suspensions of microorganisms.

Specific comments

1. References in the Title, Abstract, and Introduction and elsewhere to microbial cells and single cells may give the impression that the authors are going to describe a technique that has single cell resolution, which is of course not the case. For example: "...a novel...method that facilitates the capture, purification and interrogation of microbial cells..." (lines 18-19) and "Attempts to purify living bacterial cells..." (line 206).

We don't believe the text is misleading, but it could be changed to address concerns by adding populations as appropriate? For example, in the abstract as follows:

L19 microbial cell populations as requested

2. The "interrogation" part of the above statement is not really addressed in this paper since the only downstream analysis presented is culturing on agar plates. This is not an "interrogation of microbial cells".

The addition of antibiotics is in itself a form of interrogation of the populations of cells, however, we believe the capture and immobilization facilitates the interrogation: we have explored salt tolerance of strains, which was part of the investigation of ion exchange parameters. We believe that the capability of interrogation is implicit in the nature of the method.

I feel the text as is self-explanatory in this regard

3. The authors are correct to stress the limitations of traditional culture-based methods in microbiology. In this context, it is noticeable that the only demonstration of the efficacy of their technique is post-separation culturing.

The primary focus of the manuscript is proof of concept. Work is ongoing to develop applications, but the proof of concept we felt would be best validated using accepted technology.

It is not uncommon to use traditional methods to validate a new method.

4. Lines 40-41. "...traditional methods of microbiology for both the analytical and preparative purification of cells have become limiting". While this is undoubtedly true, I don't think the authors have made the case that their new technique overcomes the limitations.

By introducing a fluidic method that maintains cell viability and captures cell populations in order to carry out a range of experiments, is unprecedented. We have been circumspect in our claims, but if this needs expanding more proactively, text can be added here to address the concerns.

The text has been modified L42-46 and noted on the marked up MS

5. The separation of organisms from the pond sample (without a prior culturing step, if I understand correctly) (yes correct) is perhaps the most interesting aspect of this paper. I think the story would be more compelling if the authors were to do something interesting with these cultures that depends upon their prior separation. In other words, how has the separation made possible approaches that would otherwise be difficult or impossible? (And let's keep in mind that metagenomics would in principle allow assembly of the seven genomes without any separation step at all). Since the organisms are all evidently

culturable, that provides a way to separate them microbiologically and it's not clear to me what value is added by the chromatography

The separation of strains is as you say not essential for some purposes, since bioinformatics can obviate the need in many cases. Where isolates are required for further analysis, plating is also a well-established method for culturable organisms. Again, while we do not wish to speculate unduly, the capability for separation of strains without the need for downstream culturing (although this was used here as a validation) can easily be envisaged (microscopy, biochemical and genomic technologies could be readily deployed).

The text has been modified L291-296 and noted on the marked up MS

6. My sense is that the pond water sample must be quite densely populated. What is the sensitivity of this technique and how is it expected to perform with samples with lower population densities?

Good question, the limits of detection are under investigation and will form the basis of further work. Just to emphasise the focus was on demonstrating a novel methodology that retained cell viability and there was significant effort in constructing the separation device. The subsequent development of applications and fine-tuning of methods is currently in hand.

Outwith the scope of the current manuscript

7. A significant omission from this paper is any discussion of (or even recognition of the existence of) other technologies and approaches that allow for the separation of mixed microbial populations. And, except in rather vague terms, the authors do not detail specific examples of possible downstream applications.

The text has been modified throughout to recognise the omission.

Reviewer #3 (Comments for the Author):

This manuscript reports on a novel method for determining the antibiotic resistance of bacterial strains and separation of bacteria from environmental samples. The authors describe the purification of cells using ion exchange chromatography and termed the method "cell chromatography." To facilitate separation of heterogeneous populations bacterial cells the method exploits the intrinsic differences in net surface charge of the bacteria. The results of this research indicate good potential of the method to be automated and to replace traditional plate count methods in certain basic procedures employed in experimental microbiology. The manuscript is well written with clear objectives and appropriate methods for answers the research questions. The results are well interpreted with reference to pertinent published research. One main issue with the manuscript is its organization.

Thank you for the positive statements

There are instances where the results section is replete with remarks and explanations/interpretations of observations that are best suited for the discussion section. Please see the following comments and suggestions for improving the manuscript.

Line 37" "growth cycle". There are five phases (stages) of the bacterial life cycle. In this respect, growth is only observed in one phase - the exponential phase - and not in the other four life cycle phases (including the long term survival phase). Therefore, the term "growth cycle" represents a condition this is practically non-existent and should be replaced with the term "life cycle".

Agreed and amended L38

Lines 79 -80: Please state culture conditions such as incubation temperature and time of incubation. These are important to ascertain whether exponential phase or stationary phase cells were used in the experiments, and for others to accurately repeat the culture conditions.

Agreed and amended L91

Lines 132 - 133: "..but this is most likely a reflection of small differences in external adherence of cells to the surface of the tips during manipulation.." This information represents interpretation/explanation of a result. It should be addressed in the discussion section for proper organization of the manuscript.

Agreed and amended L151

Lines 165 - 172: This information highlights the importance of direct genome sequencing, and the bioprospecting opportunity offered by unculturable microbes. This also has no place in the results section of the manuscript, and should be addressed in the discussion section. Please check the results section and make similar revisions where appropriate for improved organization of the manuscript.

Agreed and amended L185-187

DTG 07/20/22

August 31, 2022

Dr. Douglas T Gjerde
Gjerde Technologies
San Jose CA95070,

Re: Spectrum02450-22R2 (Cell Chromatography: Biocompatible chromatographic separation and interrogation of microbial cells)

Dear Dr. Douglas T Gjerde:

Thank you for submitting the revision and detailed explanation of your manuscript. Feel free to improve the figures as you wish to help the readers to appreciate your work more.

Your manuscript has been accepted, and I am forwarding it to the ASM Journals Department for publication. You will be notified when your proofs are ready to be viewed.

Sincerely,

Montarop Yamabhai
Editor, Microbiology Spectrum
